# Synaptic input variation enhances rate coding at the expense of temporal precision in cochlear nucleus neurons

Chunjian Wang[1,2], Go Ashida[3,4], Christian Keine[1,2], Ivan Milenkovic[1,2,4]*

1 Department for Human Medicine, Carl von Ossietzky Universität Oldenburg, Oldenburg, Germany, 2 Research Center Neurosensory Science, Carl von Ossietzky Universität Oldenburg, Oldenburg, Germany, 3 Department for Neuroscience, Carl von Ossietzky Universität Oldenburg, Oldenburg, Germany, 4 Cluster of Excellence "Hearing4all", Carl von Ossietzky Universität Oldenburg, Oldenburg, Germany

* ivan.milenkovic@uol.de

## Abstract

Synaptic convergence is fundamental to neuronal circuit function, underpinning computations such as coincidence detection and signal integration. Across sensory systems, convergence architecture and synaptic input strengths are key for extracting stimulus features and processing of sensory information. In the cochlear nucleus, globular bushy cells (GBCs) receive convergent inputs from multiple auditory nerve fibers via large endbulb of Held terminals. While these inputs vary considerably in size, even among those targeting the same cell, the functional consequences of this variation for sound encoding remain unclear. Here, we investigated how synaptic input variation shapes sound encoding in GBCs of Mongolian gerbils using in vitro conductance-clamp recordings and computational modeling. By simulating synaptic inputs with variable strength distributions, we found that increasing input variation enhances rate coding at the expense of temporal precision. These findings suggest that endbulb strength heterogeneity allows the GBC population to operate along a functional continuum, generating diverse information streams to downstream targets.

## Introduction

Neuronal circuits process information through intricate patterns of synaptic communication. A fundamental principle governing these circuits is synaptic convergence, where individual neurons integrate signals from multiple presynaptic sources. The degree of convergence varies widely across cell types and brain regions, reflecting adaptations to circuit-specific computational demands.

This principle is particularly evident in sensory systems, where parallel pathways often specialize in extracting distinct stimulus features [1–5]. For example, the auditory brainstem exhibits a broad spectrum of synaptic convergence. At one extreme,

**Data availability statement:** All numerical data and model code used to generate figures are available at https://doi.org/10.17632/2tb-kzdpjy6 under the terms of Creative Commons Attribution 4.0 License (CC BY 4.0).

**Funding:** This work was funded by the Deutsche Forschungsgemeinschaft (DFG, German Research Foundation, www.dfg.de) under Germany's Excellence Strategy – EXC 2177/1 - Project ID 390895286 to GA, the Open Access Publication Funding (DFG), and intramural funding to CK (Carl von Ossietzky Universität Oldenburg, www.uol.de; FP 2024-099). The funders had no role in study design, data collection and analysis, decision to publish, or preparation of the manuscript.

**Competing interests:** The authors have declared that no competing interests exist.

**Abbreviations:** AM, amplitude-modulated; AN, auditory nerve; AP, action potential; AVCN, anteroventral cochlear nucleus; CF, characteristic frequency; CI, correlation index; CN, cochlear nucleus; GBC, globular bushy cell; LSO, lateral superior olive; MNTB, medial nucleus of the trapezoid body; MSO, medial superior olive; PBS, phosphate-buffered saline; PSTH, peri-stimulus time histogram; PVCN, posteroventral cochlear nucleus; SAM, sinusoidal amplitude-modulated; SGN, spiral ganglion neuron; SOC, superior olivary complex; SPN, superior paraolivary nucleus; SR, spontaneous rate.

neurons in the medial nucleus of the trapezoid body (MNTB) are driven by a single large axosomatic terminal, the calyx of Held, which ensures fast and reliable signal transmission [6–11]. In contrast, octopus and D-stellate cells in the cochlear nucleus (CN) integrate signals from dozens of auditory nerve (AN) fibers, responding selectively to the coincident firing of their inputs [12–16]. Even within a single cell type, convergence can differ by orders of magnitude: primate retinal ganglion cells in the fovea receive input from only a few bipolar cells, whereas those in the periphery integrate signals from hundreds of inputs [17–19]. In the nucleus magnocellularis, the avian CN homolog, synaptic convergence is tightly aligned with functional demands: neurons tuned to low frequencies receive numerous small inputs, whereas high-frequency neurons are contacted by fewer, larger terminals to optimize temporal processing [20,21]. Together, these examples illustrate how convergence patterns are finely tuned to the functional requirements of a circuit.

Beyond the sheer number of inputs, variation in synaptic strength among converging inputs represents another critical, yet less understood, factor in neuronal circuit function [22–26]. Across diverse brain regions, including the cerebellum, thalamus, and cortex, some neurons integrate many weak subthreshold inputs, while others are dominated by one or two suprathreshold "driver" inputs alongside multiple weaker ones [22,23,26–28]. Understanding how this variation in synaptic strength shapes neuronal processing is essential for deciphering the computational principles of sensory signaling.

In this study, we investigated globular bushy cells (GBCs) in the mammalian CN as a model system to explore the functional impact of synaptic input variation. Located near the AN root in the anteroventral (AVCN) and posteroventral cochlear nucleus (PVCN) [29,30], GBCs receive excitatory inputs from multiple AN fibers via large presynaptic terminals known as endbulbs of Held. The number of inputs ranges from 5 to 12 in mice [14,31] to over 20 in cats [32–35]. Functionally, GBCs are essential for high-fidelity temporal processing of auditory signals and often exhibit greater temporal precision than their AN inputs [1,36,37]. Recent morphological analyses have revealed considerable heterogeneity in the size of endbulbs contacting individual GBCs. While some GBCs receive inputs from many similarly sized, subthreshold endbulbs, others are innervated by a few large, suprathreshold endbulbs alongside smaller ones [31,38]. Despite this documented morphological heterogeneity, the computational implications for GBC sound processing remain unclear.

To address this knowledge gap, we combined in vitro whole-cell conductance-clamp recordings from GBCs in Mongolian gerbils with computational modeling. By simulating varying degrees of heterogeneity among AN inputs, we assessed the impact of this variation on GBC firing rates and temporal precision. Our results demonstrate that the degree of input variation critically shapes GBC output. Specifically, increasing variation elevates firing rates at the cost of temporal precision in encoding stimulus fine structure. These findings suggest that morphological diversity among endbulb inputs is a key parameter determining GBC encoding strategy, thereby regulating the auditory cues delivered to downstream sound localization circuits.

## Results

### Synaptic input variation enhances GBC firing but reduces temporal precision

To study the impact of synaptic input variation, we targeted GBCs based on their well-established anatomical and electrophysiological characteristics. Located in the caudal AVCN near the AN root and PVCN [29], GBCs are characterized by round-to-oval cell bodies with few short dendrites (Fig 1A1). Electrophysiologically, GBCs fire one or a few action potentials (APs) in response to suprathreshold current injection and exhibit low input resistance ($45.5 \pm 18$ MΩ) and a fast membrane time constant ($1.1 \pm 0.6$ ms, $n = 70$) at rest (Fig 1A2). In response to low-frequency sounds, GBCs sustain firing rates up to several hundred spikes per second, with APs temporally aligned to the stimulus fine-structure [36,37,39].

Using an established model of the auditory periphery [40], we simulated independent spike trains for 10 convergent AN fibers responding to 200-ms pure tones (10 repetitions) at frequencies of 350 Hz, 1,000 Hz, and 3,500 Hz. These frequencies were selected to capture the range where AN fibers exhibit robust phase-locking (350 and 1,000 Hz) up to the limit where phase-locking declines in rodents (3,500 Hz) [37,39]. The resulting AN spike trains were then convolved with an experimentally derived EPSC waveform to create conductance trains, which were injected into recorded GBCs. By systematically varying the synaptic input strength distribution across these 10 simulated AN fibers, we defined three distinct conditions: "no variation" (all inputs of equal strength: coefficient of variation $v = 0$), "medium variation" ($v = 0.4$), and "high variation" ($v = 1$) (Fig 1B). In the "high variation" condition, the largest input was near or above spike threshold, while for the "no" and "medium" variation conditions, all inputs remained subthreshold.

First, we analyzed how input variation impacted firing rates and observed a frequency-dependent effect. At stimulation frequencies of 1,000 and 3,500 Hz, increasing input variation elevated average GBC firing rates (1 kHz: "no variation" = $135 \pm 49$ Hz versus "high variation" = $175 \pm 35$ Hz, $p = 7.9e{-6}$, $n = 15$, Fig 1C1, Table 1). In contrast, during 350 Hz stimulation, firing rates decreased slightly under "high variation" ("no variation" = $251 \pm 53$ Hz versus "high variation" = $234 \pm 40$ Hz, $p = 0.01$). These changes were uncorrelated with the cells' input resistance ($r_s = -0.13$, $p = 0.68$) or membrane time constant ($r_s = 0.13$, $p = 0.67$).

Next, we assessed temporal precision using vector strength (VS) [41]. Simulated AN inputs exhibited VS values of 0.75 at 350 Hz and 0.77 at 1,000 Hz, consistent with in vivo observations [42–45]. Postsynaptically, GBCs showed robust phase-locking at 350 Hz and 1,000 Hz, but not at 3,500 Hz, aligning with previous data [37]. Notably, increasing input variation significantly reduced phase-locking, particularly at 350 Hz (main effect variation: $p = 3.7e{-6}$; 350 Hz: "no variation" = $0.86 \pm 0.02$ versus "high variation" = $0.81 \pm 0.01$, $p = 3.3e{-8}$, Fig 1C2). Similar to firing rates, the reduction in VS was independent of input resistance ($r_s = -0.31$, $p = 0.3$) or membrane time constant ($r_s = -0.16$, $p = 0.59$). Together, these results indicate a trade-off whereby higher input variation increases firing rates at the cost of low-frequency phase-locking.

### Effects of input variation are well reproduced in a coincidence-counting model

To determine whether these experimental observations can be generalized and arose primarily from input statistics, we employed a computational GBC model, originally developed for cats and adapted for gerbils [4]. In this model, an output spike is generated when the summed inputs exceed an adaptive threshold (see Methods). We drove the model with simulated AN spike trains identical to those used in our in vitro experiments, testing a wide range of input variations, including the "no," "medium," and "high" variations from our conductance-clamp recordings (Fig 2A). The model faithfully reproduced the key experimental findings of increased firing rates (Fig 2B) and reduced VS with higher input variation (Fig 2C). This suggests that the interaction between synaptic input statistics and a simplified cellular integration mechanism is sufficient to replicate the experimentally observed phenomena.

### Higher synaptic input variation impacts AP latency and increases temporal jitter

Given that increased synaptic input variation enhances overall firing rates (Fig 1C1), we next investigated its impact on the onset and sustained phases of GBC responses. To isolate these epochs, we constructed peristimulus time histograms

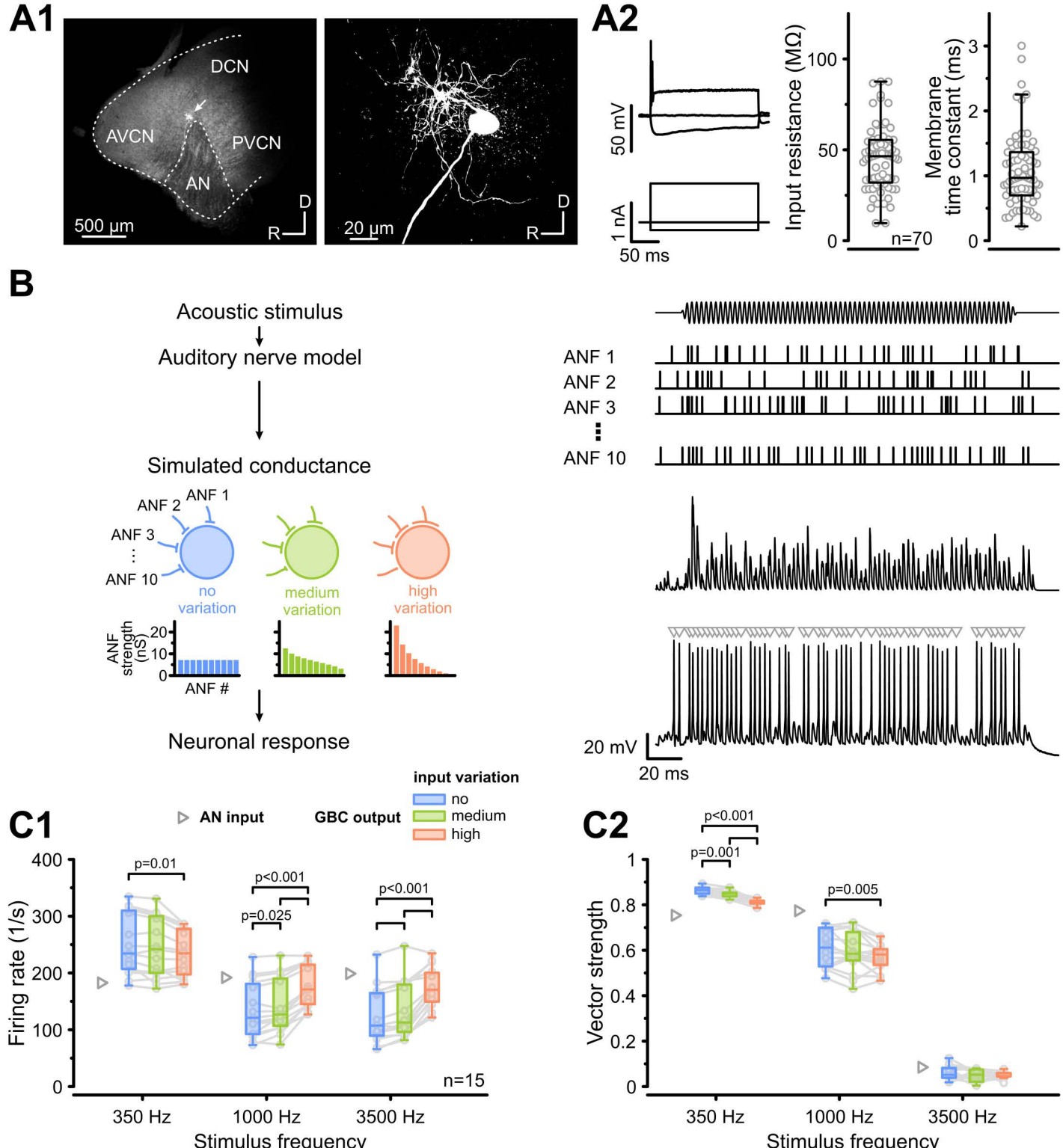

**Fig 1. Synaptic input variation increases firing rates at the expense of temporal precision. (A)** GBC identification based on morphological and electrophysiological features. **(A1)** Left: Low-magnification image of cochlear nucleus parasagittal slice containing a biocytin-filled GBC near the AN root (arrow). AVCN, anteroventral cochlear nucleus; PVCN, posteroventral cochlear nucleus; DCN, dorsal cochlear nucleus; AN, auditory nerve. Right:

High-magnification image of the same GBC, displaying its characteristic round soma and short dendrites. **(A2)** Left: Electrophysiological signature of a GBC (top) responding with one or a few APs at the onset of suprathreshold depolarizing current (bottom) and voltage sag in response to hyperpolarizing current injection. Right: Population data ($n = 70$) of GBC input resistance and membrane time constant. **(B)** Schematic of the experimental paradigm: Acoustic stimuli (top) were processed by an auditory nerve model to generate spike trains for 10 AN fibers (ANF1-10). These spike trains were convolved with the EPSC waveform to generate conductance templates with varying synaptic strengths, ranging from "no variation" (blue) to "high variation" (orange), and injected into GBCs. Histograms illustrate the input distributions for the conditions used. The GBC response (bottom right) was recorded, and AP number and timing (gray triangles) were analyzed. **(C1)** Population data ($n = 15$) of the average GBC firing rate in response to pure-tone stimuli (350, 1,000, and 3,500 Hz) under three input conditions (blue: "no variation," green: "medium variation," orange: "high variation"). Gray triangles: AN input. Firing rates increased with input variation for 1,000 and 3,500 Hz stimulation. **(C2)** Vector strength, as a measure of temporal precision, decreased with input variation, particularly at lower frequencies. Boxplots show median and interquartile range (IQR, whiskers: 1.5 × IQR). Individual cells are represented by gray circles. *P*-values: 2-way RM ANOVA with Bonferroni post-hoc tests ($n = 15$). The data underlying this Figure are available at https://doi.org/10.17632/2tbkzdpjy6.

(PSTH) from responses to a 3,500 Hz pure-tone, a frequency selected to minimize confounding phase-locking effects. Consistent with in vivo recordings [1,46–48], simulated AN inputs and GBC responses both displayed primary-like PSTHs characterized by a sharp onset peak followed by adaptation to a lower sustained rate (Fig 3A). Additionally, GBC PSTHs displayed the characteristic "notch" immediately following the onset peak. Quantitative analysis confirmed that input variation affected response epochs differently. While "high variation" significantly reduced onset spike probability ("no variation" = 0.97 ± 0.02 versus "high variation" = 0.94 ± 0.02, $p = 7.3e-5$, $n = 17$, Fig 3B2) it increased both sustained firing rates and spontaneous activity (sustained: "no variation" = 72 ± 27 Hz versus "high variation" = 131 ± 28 Hz, $p = 3.2e-11$, Fig 3B3; spontaneous: "no variation" = 25 ± 14 Hz versus "high variation" = 65 ± 15 Hz, $p = 4.9e-11$, Fig 3B4).

To probe the timing of the response onset, we analyzed the minimum first-spike latency ($FSL_{min}$) and jitter. Because high spontaneous rates (SRs) can lead to an apparent decrease in latency, we used a method that accounts for differences in baseline firing [49]. This revealed a nuanced result, with $FSL_{min}$ slightly but significantly prolonged in the "high variation" condition ("no variation" = 2.83 ± 0.11 ms versus "high variation" = 2.88 ± 0.09 ms, $p = 4.5e-4$, Fig 3B5). While first-spike jitter in GBCs was generally lower compared to their AN input, it significantly increased with higher input variation ("no variation" = 0.22 ± 0.02 ms versus "high variation" = 0.27 ± 0.02 ms, $p = 1.6e-9$, Fig 3B6). The computational model replicated those findings (Fig 3C), reproducing the characteristic PSTH shapes (Fig 3A3) and the gradual change of firing rates with input variation. The model offered further insight into SRs, revealing a sharp increase at input variations of 0.7–0.8 (Fig 3C4), corresponding to the transition point where the largest input became suprathreshold. Although the model also captured the increases in $FSL_{min}$ and first-spike jitter (Fig 3C5 and 3C6), the simulated distributions were narrower than the experimental data, likely reflecting the absence of biological noise sources (e.g., channel and thermal fluctuations). In summary, both experimental and modeling results demonstrated that input variation differentially sculpts GBC responses by enhancing sustained firing rates while simultaneously degrading both the reliability and temporal precision of the response onset.

## Input variation modulates GBC encoding of amplitude-modulated sounds

The results from pure-tone stimulation suggest a trade-off where low input variation favors high temporal precision, while high input variation enhances firing rates. To further explore this phenomenon in the context of more complex, naturalistic stimuli, we examined GBC responses to sinusoidal amplitude-modulated (SAM) sounds, which capture the temporal fluctuations present in speech and environmental sounds [50–52]. Effective SAM encoding requires both precise spike timing relative to the modulation cycle (temporal code) and the ability of firing rates to track changes in sound level (rate code). GBCs are known to effectively follow rapid amplitude modulations [53–55].

We simulated the responses of AN fibers to a 13 kHz tonal carrier, 100% amplitude-modulated at frequencies of 200, 600, and 2,000 Hz. Consistent with previous in vivo observations [56,57], AN responses exhibited envelope phase-locking for 200 and 600 Hz. Injecting these conductance trains into GBCs under the "no", "medium", and "high variation"

**Table 1. Summary of statistical analysis for gamma-distributed input variations.**

| | | | Variation | | | | |
| --- | --- | --- | --- | --- | --- | --- | --- |
| | | Stimulus | No | Medium | High | *p*-value RM ANOVA | Post-hoc comparisons |
| Fig 1 (*n*=15) | Firing Rate (Hz) | 350 Hz | 251±53 | 245±53 | 234±40 | Variation: $p=4.6e{-}5$ | No vs. medium: $p=0.1$<br>No vs. high: $p=0.01$<br>Medium vs. high: $p=0.05$ |
| | | 1,000 Hz | 135±49 | 143±47 | 175±35 | | No vs. medium: $p=0.03$<br>No vs. high: $p=7.9e{-}6$<br>Medium vs. high: $p=6.6e{-}6$ |
| | | 3,500 Hz | 122±48 | 134±49 | 174±34 | | No vs. medium: $p=5.5e{-}5$<br>No vs. high: $p=8.9e{-}7$<br>Medium vs. high: $p=1.4e{-}5$ |
| | Vector Strength | 350 Hz | 0.86±0.02 | 0.85±0.01 | 0.81±0.01 | Variation: $p=3.7e{-}6$ | No vs. medium: $p=0.001$<br>No vs. high: $p=3.3e{-}8$<br>Medium vs. high: $p=4.3e{-}8$ |
| | | 1,000 Hz | 0.61±0.08 | 0.59±0.09 | 0.57±0.06 | | No vs. medium: $p=0.43$<br>No vs. high: $p=0.005$<br>Medium vs. high: $p=0.29$ |
| | | 3,500 Hz | 0.06±0.03 | 0.05±0.03 | 0.05±0.02 | | No vs. medium: $p=0.57$<br>No vs. high: $p>0.99$<br>Medium vs. high: $p>0.99$ |
| Fig 3 (*n*=15) | Onset probability | 3,500 Hz | 0.97±0.02 | 0.96±0.02 | 0.94±0.02 | $p=3.5e{-}6$ | No vs. medium: $p=0.42$<br>No vs. high: $p=7.3e{-}5$<br>Medium vs. high: $p=4e{-}4$ |
| | Sustained firing rate (Hz) | 3,500 Hz | 71.7±26.8 | 86.6±29.7 | 130.8±28.2 | $p=8.7e{-}16$ | No vs. medium: $p=5.8e{-}5$<br>No vs. high: $p=3.2e{-}11$<br>Medium vs. high: $p=2.7e{-}12$ |
| | Spontaneous firing rate (Hz) | 3,500 Hz | 24.8±14.3 | 34.8±17.8 | 65±14.7 | $p=2.4e{-}17$ | No vs. medium: $p=9e{-}4$<br>No vs. high: $p=4.9e{-}11$<br>Medium vs. high: $p=1.8e{-}9$ |
| | Minimum first-spike latency (ms) | 3,500 Hz | 2.83±0.11 | 2.82±0.1 | 2.88±0.09 | $p=8.8e{-}8$ | No vs. medium: $p=0.05$<br>No vs. high: $p=4.5e{-}4$<br>Medium vs. high: $p=1.7e{-}7$ |
| | First-spike jitter (ms) | 3,500 Hz | 0.22±0.02 | 0.24±0.02 | 0.27±0.02 | $p=7.3e{-}13$ | No vs. medium: $p=8e{-}4$<br>No vs. high: $p=1.6e{-}9$<br>Medium vs. high: $p=5e{-}6$ |
| Fig 4 (*n*=15) | Firing Rate (Hz) | 200 Hz | 130±36 | 131±33 | 145±23 | Variation: $p=3e{-}10$ | No vs. medium: $p>0.99$<br>No vs. high: $p=0.01$<br>Medium vs. high: $p=0.002$ |
| | | 600 Hz | 67±50 | 77±50 | 115±38 | | No vs. medium: $p=3e{-}4$<br>No vs. high: $p=1.47e{-}7$<br>Medium vs. high: $p=3.5e{-}6$ |
| | | 2,000 Hz | 65±50 | 71±46 | 111±43 | | No vs. medium: $p=0.12$<br>No vs. high: $p=7.8e{-}9$<br>Medium vs. high: $p=1.4e{-}9$ |
| | Vector Strength | 200 Hz | 0.84±0.03 | 0.83±0.02 | 0.76±0.01 | Variation: $p=5.7e{-}14$ | No vs. medium: $p=0.006$<br>No vs. high: $p=9.3e{-}10$<br>Medium vs. high: $p=1.6e{-}12$ |
| | | 600 Hz | 0.51±0.06 | 0.48±0.05 | 0.37±0.03 | | No vs. medium: $p=2.2e{-}6$<br>No vs. high: $p=7.2e{-}10$<br>Medium vs. high: $p=9.6e{-}10$ |
| | | 2,000 Hz | 0.01±0.01 | 0.01±0.01 | 0.01±0 | | No vs. medium: $p=0.84$<br>No vs. high: $p>0.99$<br>Medium vs. high: $p=0.1$ |

*(Continued)*

**Table 1.** (Continued)

| | Stimulus | Variation | | | *p*-value RM ANOVA | Post-hoc comparisons |
|---|---|---|---|---|---|---|
| | | **No** | **Medium** | **High** | | |
| **Correlation Index** | **200** Hz | 3.06±0.19 | 2.93±0.1 | 2.39±0.06 | Variation: *p*=5.9e−13 | No vs. medium: *p*=4e−4<br>No vs. high: *p*=3.7e−10<br>Medium vs. high: *p*=1.3e−12 |
| | **600** Hz | 1.53±0.12 | 1.46±0.1 | 1.26±0.04 | | No vs. medium: *p*=6.2e−6<br>No vs. high: *p*=3e−8<br>Medium vs. high: *p*=2.6e−8 |
| | **2,000** Hz | 1.01±0.02 | 1.01±0.02 | 1±0.01 | | No vs. medium: *p*>0.99<br>No vs. high: *p*=0.5<br>Medium vs. high: *p*=0.06 |
| **Entrainment Index** | **200** Hz | 0.6±0.17 | 0.62±0.17 | 0.68±0.1 | Variation: *p*=4.7e−6 | No vs. medium: *p*=0.87<br>No vs. high: *p*=0.01<br>Medium vs. high: *p*=0.04 |
| | **600** Hz | 0.01±0.01 | 0.01±0.03 | 0.04±0.05 | | No vs. medium: *p*=0.36<br>No vs. high: *p*=0.005<br>Medium vs. high: *p*=0.001 |
| | **2,000** Hz | 0±0 | 0±0 | 0±0 | | No vs. medium: *p*=0.44<br>No vs. high: *p*=0.64<br>Medium vs. high: *p*>0.99 |
| **Corr$_{Norm}$** | **200** Hz | 0.87±0.02 | 0.88±0.01 | 0.93±0.01 | Variation: *p*=1.8e−10 | No vs. medium: *p*=0.004<br>No vs. high: *p*=2.5e−10<br>Medium vs. high: *p*=6.3e−11 |
| | **600** Hz | 0.99±0.01 | 0.99±0 | 0.98±0.01 | | No vs. medium: *p*=0.027<br>No vs. high: *p*=0.5<br>Medium vs. high: *p*=0.003 |
| | **2,000** Hz | 0.82±0.01 | 0.82±0.01 | 0.82±0 | | No vs. medium: *p*>0.99<br>No vs. high: *p*=0.42<br>Medium vs. high: *p*=0.26 |

conditions yielded robustly modulated firing rates (Fig 4A2), a result faithfully reproduced by the GBC model (Fig 4A3). In both the experimental data and the model, responses appeared sharper and more temporally restricted under the "no variation" condition compared with the "high variation" condition.

Consistent with pure-tone data, overall firing rates during SAM stimulation increased significantly with input variation across all modulation frequencies (main effect of variation: *p*=3e−10; 600 Hz: "no variation"=67±50 Hz versus "high variation"=115±38 Hz, *p*=1.5e−7, *n*=15, Fig 4B1). Next, we assessed temporal coding fidelity using two complementary metrics: VS and correlation index (CI). The VS indicated robust phase-locking for 200 and 600 Hz but decreased significantly with higher input variation (main effect of variation: *p*=5.7e−14, 600 Hz: "no variation'=0.51±0.06 versus "high variation"=0.37±0.03, *p*=7.2e−10, Fig 4B2). This is consistent with the reduced VS observed during pure-tone stimulation, indicating weaker encoding of the modulation fine-structure. Similarly, the CI, which quantifies spike time precision across stimulus repetitions [58,59], decreased significantly with higher input variation (main effect of variation: *p*=5.9e−13; 600 Hz: "no variation"=1.53±0.12 versus "high variation"=1.26±0.04, *p*=3e−8, Fig 4B3). Together, these measures indicate that increased input variation reduces both phase-locking to the modulation cycle and spike time consistency across trials. In contrast, entrainment index increased significantly with input variation at low modulation frequencies (main effect of variation: *p*=4.7e−6, 200 Hz: "no variation"=0.6±0.17 versus "high variation"=0.68±0.1, *p*=0.01, Fig 4B4). At 200 Hz, the distributions of the measured and simulated entrainment index were broader than those of the VS (Fig 4B2), likely because entrainment depends on firing rate, which was broadly distributed (Fig 4B1), whereas phase-locking (measured by VS) is independent of firing rate.

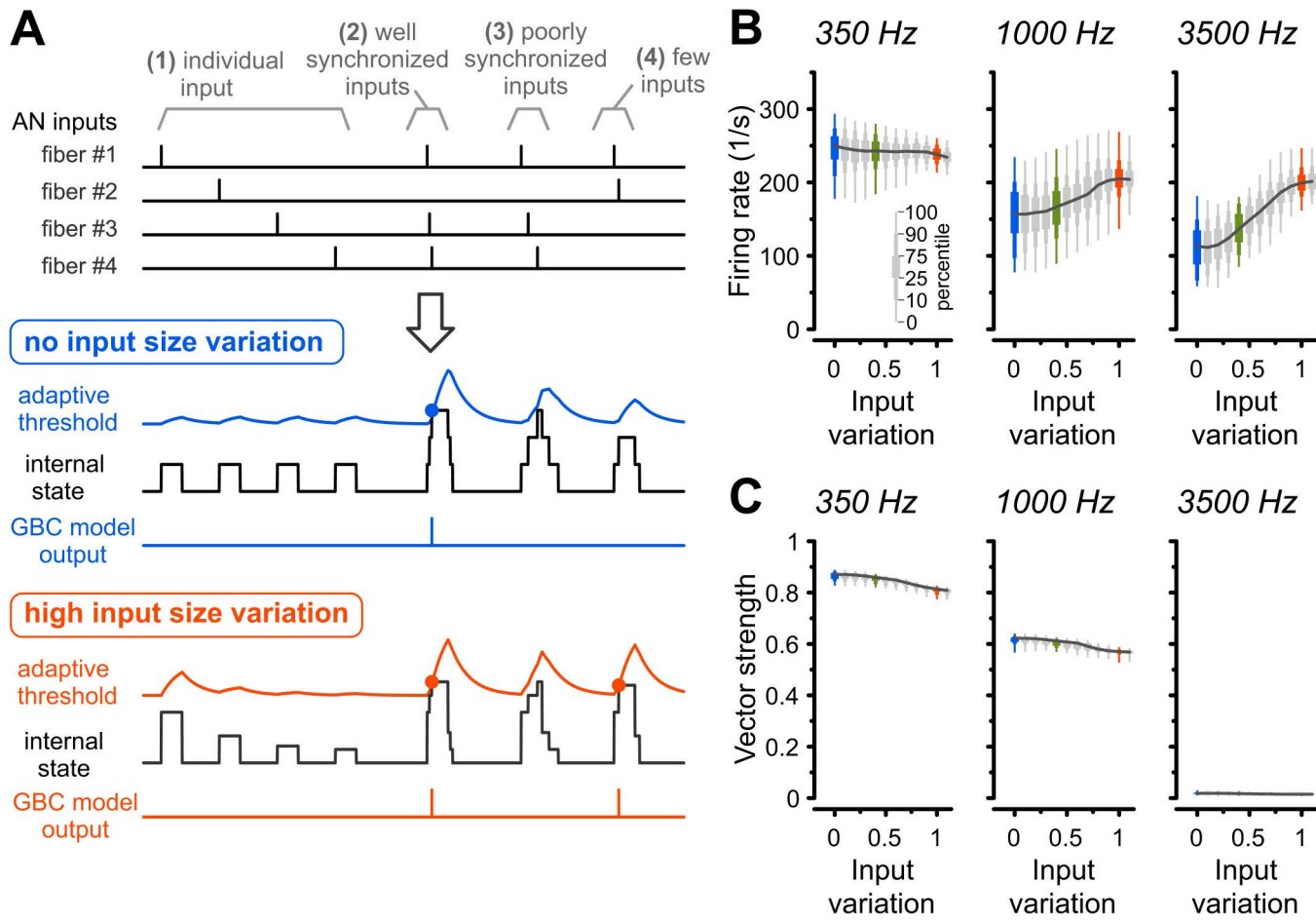

**Fig 2. Adaptive coincidence-counting model reproduces experimental data across a wide range of input variations. (A)** Schematic of GBC model. Top: Example spike trains of four AN fibers highlighting different spike synchrony. Middle: Model responses to AN fiber inputs with no (blue) and high (orange) input variations. The adaptive threshold changes dynamically according to the recent history of summed inputs (internal state). GBC model output spikes are generated when the summed inputs exceed this threshold (blue and orange circles). Note that in the "high variation" condition, strong individual inputs can generate GBC spikes with little dependence on coincidence. **(B)** Average firing rates for 313 GBC model instances in response to pure-tone stimuli (350, 1,000, and 3,500 Hz). Colored data indicate input variations used in the in vitro experiments (blue: "no variation," green: "medium variation," orange: "high variation"). The model reproduces the experimental trend of higher firing rates for larger input variations at high frequencies. **(C)** Vector strength reduced with increasing input variation, mirroring the experimental results. Box plots show the range of values from all 313 model instances with thick, medium, and thin boxes indicating 25–75, 10–90, and 0–100 percentiles, respectively. The data underlying this Figure are available at https://doi.org/10.17632/2tbkzdpjy6.

Next, we investigated how well GBC firing rates tracked the stimulus envelope, reflecting their rate-coding capacity. Normalized cross-correlation ($Corr_{Norm}$) revealed that firing rates tracked the stimulus envelope more effectively under "high variation" (main effect of variation: $p = 1.8e{-}10$; 200 Hz: "no variation" $= 0.87 \pm 0.02$ versus "high variation" $= 0.93 \pm 0.01$, $p = 2.5e{-}10$, Fig 4B5). In summary, during SAM stimulation, increased input variation shifts the GBC coding strategy by degrading fine-structure temporal precision (Figs 1C2 and 2C) while simultaneously improving envelope representation through firing rate modulation. This suggests that GBCs receiving different convergent patterns of AN inputs might be differentially suited to either favor precise temporal representation, requiring highly coincident inputs, or emphasize envelope tracking via firing rate modulation and generally higher firing rates [60–62].

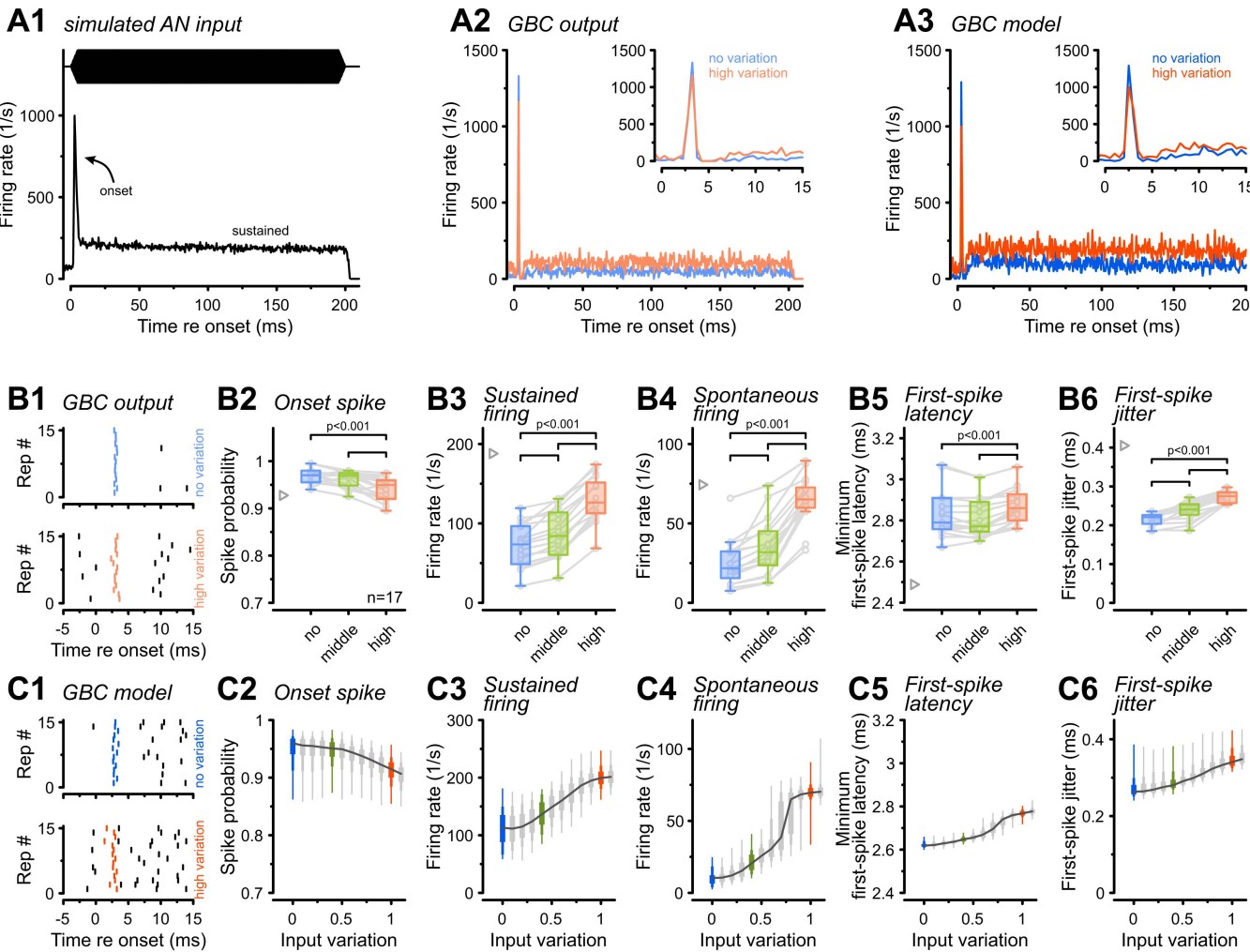

**Fig 3. Synaptic input variation increases sustained firing rates but reduces onset spike-time precision. (A1)** Peristimulus time histogram (PSTH) of the average AN input responding to a 3,500 Hz pure-tone. Note the sharp onset followed by adaptation to a sustained rate. **(A2)** Representative GBC output PSTH for the input in A1 under "no variation" (blue) and "high variation" (orange). **(A3)** Output of the GBC model in response to the same conditions, mirroring experimental data. Insets in A2–A3 show a magnified view of the response onset, highlighting the reduced onset peak under the "high variation" condition. **(B)** Experimental population data ($n = 17$) for different input variations (blue: "no variation," green: "medium variation," orange: "high variation"). **(B1)** Raster plot (15 repetitions) of the first spike within 1.5 ms of stimulus onset during "no variation" (top, blue) and "high variation" (bottom, orange) conditions. Note the slightly earlier but more variable first spike in the "high variation" condition. **(B2)** Onset spike probability was significantly reduced with input variation. **(B3–B4)** sustained firing rates (50–200 ms) and spontaneous firing rates increased with input variation. Gray triangles: mean AN input. **(B5–B6)** Minimum first-spike latency and first-spike jitter increased with input variation. **(C1–C6)** Output of the GBC model quantifying the same parameters as for the experimental data in B1–B6. Colored box plots indicate the input variations used in the in vitro experiments. In (B) box plots show median, IQR (whiskers: 1.5 × IQR) with individual cells shown as gray circles. In (C) box plots for modeling data show the range of values from all 313 model instances with percentiles (thick: 25–75, medium: 10–90, thin: 0–100). *P*-values: 2-way RM ANOVA with Bonferroni post-hoc tests ($n = 17$). The data underlying this Figure are available at https://doi.org/10.17632/2tbkzdpjy6.

## Impact of input variation is consistent with morphologically plausible input sizes

Our results suggest that input variation shifts the operational focus of GBCs between temporal precision (coincidence detection) and firing rate encoding. To determine if this principle arises directly from the known anatomical diversity, we next generated synaptic input variations constrained by morphological measurements of mouse GBCs [31]. From these

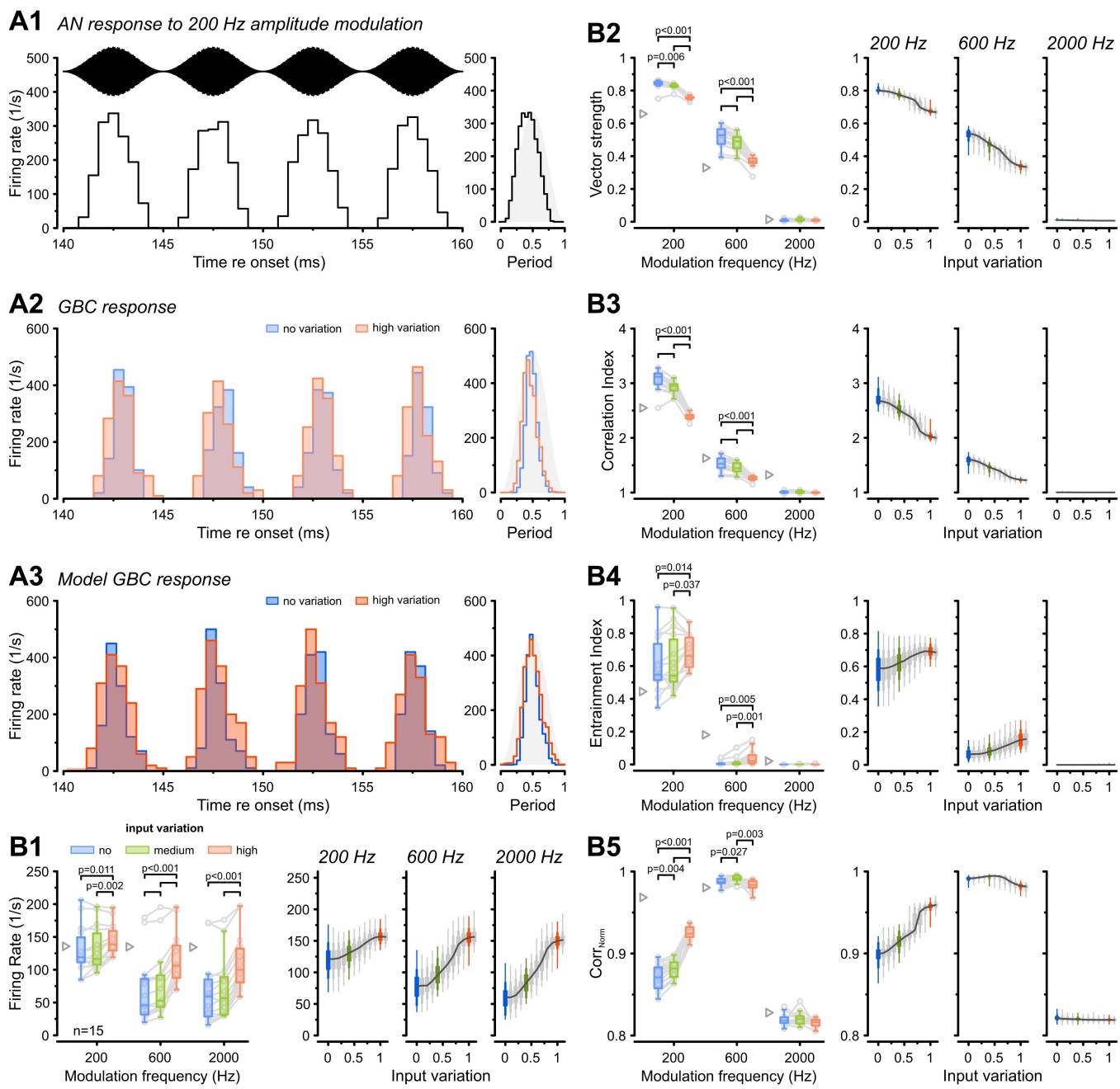

**Fig 4. Synaptic input variation shapes GBC encoding of amplitude-modulated sounds. (A1)** Simulated AN fiber response to a 200 Hz amplitude-modulated sound (top). The AN firing rate was robustly modulated by the stimulus envelope. Note that the stimulus was aligned with the AN firing rate modulation for display purposes. **(A2)** PSTHs of a representative GBC responding to input in A1. Note the broader peaks in the "high variation" (orange) condition compared to "no variation" (blue). **(A3)** GBC model response to the same input, reproducing the general PSTH shape, including broader peaks in the "high variation" condition. **(B)** Population data ($n = 15$) for modulation frequencies of 200, 600, and 2,000 Hz. Left: Experimental data; Right: Model responses. Gray triangles: mean AN input. **(B1)** Average firing rates generally increased with input variation. **(B2–B3)** Vector strength and correlation index decreased significantly with input variation. **(B4–B5)** Entrainment index and normalized cross-correlation ($Corr_{Norm}$) increased with input variation at low modulation frequencies but remained unchanged at high frequencies. Box plots containing experimental data show median and IQR (whiskers: 1.5 × IQR). Individual cells shown as gray circles. Box plots for modeling data show the range of values from all 313 model instances with percentiles (thick: 25–75, medium: 10–90, thin: 0–100). *P*-values: 2-way RM ANOVA with Bonferroni post-hoc tests ($n = 15$). The data underlying this Figure are available at https://doi.org/10.17632/2tbkzdpjy6.

measurements, we derived two conditions approximating the variance of our previous "medium" ($v \approx 0.33$, $morph_{MED}$) and "high" ($v \approx 0.8$, $morph_{HIGH}$) conditions.

GBC responses to pure tones under morphologically constrained input variations were in line with our previous observations. Consistent with the shift towards rate coding, the $morph_{HIGH}$ condition yielded significantly higher firing rates at 1,000 and 3,500 Hz stimulation frequencies (1,000 Hz: $morph_{MED}$ = 94 ± 28 Hz versus $morph_{HIGH}$ = 121 ± 33 Hz, $p$ = 3.8e−9, $n$ = 17, Fig 5A, Table 2). Concurrently, VS at 350 Hz and 1,000 Hz was significantly reduced under high input variation (1,000 Hz: $morph_{MED}$ = 0.65 ± 0.04 versus $morph_{HIGH}$ = 0.62 ± 0.05, $p$ = 5.3e−5, Fig 5B). Analysis of onset responses revealed that $morph_{HIGH}$ prolonged $FSL_{min}$ (3,500 Hz: $morph_{MED}$ = 2.84 ± 0.07 ms versus $morph_{HIGH}$ = 2.91 ± 0.07, $p$ = 5.3e−7, Fig 5C) and increased temporal jitter (3,500 Hz: $morph_{MED}$ = 0.24 ± 0.01 ms versus $morph_{HIGH}$ = 0.29 ± 0.01, $p$ = 7.7e−10, Fig 5D). These results were reproduced in the computational model and mirror the patterns observed with gamma-distributed variations (Figs 1 and 3). This suggests that the modulation of GBC output is a robust functional consequence of the morphological heterogeneity of endbulb inputs.

### Morphologically plausible input variation impacts encoding of amplitude-modulated sounds

To confirm the impact of realistic input variations on the encoding of amplitude-modulated sounds, we repeated the SAM stimulation experiments using the two morphologically constrained input variations. Consistent with the results from gamma-distributed variations (Fig 4), the morphologically constrained inputs presented a trade-off between firing rate and temporal precision. Higher input variation elevated overall firing rates (600 Hz: $morph_{MED}$ = 96 ± 38 Hz versus $morph_{HIGH}$ = 125 ± 34 Hz, $p$ = 7.4e−9, $n$ = 19, Fig 6A), whereas VS and CI were concomitantly reduced (VS at 600 Hz: $morph_{MED}$ = 0.46 ± 0.04 versus $morph_{HIGH}$ = 0.37 ± 0.02, $p$ = 8.1e−12; CI at 600 Hz: $morph_{MED}$ = 1.43 ± 0.07 versus $morph_{HIGH}$ = 1.26 ± 0.03, $p$ = 1.2e−10, Fig 6B and 6C). However, rate-coding fidelity ($Corr_{Norm}$) was enhanced at 200 Hz modulation frequency under the $morph_{HIGH}$ condition (200 Hz: $morph_{MED}$ = 0.89 ± 0.01 versus $morph_{HIGH}$ = 0.93 ± 0.01 Hz, $p$ = 2.9e−17, Fig 6D). These findings reinforce the conclusion that the degree of morphological heterogeneity among endbulb inputs critically shapes the balance between temporal and rate-coding strategies during the processing of both simple and complex sounds.

## Discussion

The heterogeneity of synaptic inputs converging onto neurons is a fundamental feature of neural circuit design. The strength and variation of these inputs profoundly shape neuronal output, thereby determining a neuron's computational role within its circuit. In this study, we investigated how such input strength variation affects sound information coding in GBCs which receive multiple, variably sized endbulb inputs from the AN. Our findings demonstrate that the degree of variation among these endbulb inputs critically determines GBC firing characteristics. Specifically, the results reveal a functional trade-off: increased input variation enhances overall firing rates and improves the encoding of stimulus amplitude modulation, yet reduces temporal precision in representing the stimulus fine-structure and increases onset spike jitter. The modeling results, which extend the experimental observations by testing a wider range of input variations, demonstrate that this functional transition occurs continuously. This suggests that morphological variation in synaptic strength enables GBCs to operate along a functional continuum, tuned by their specific input configurations.

### Computational mechanisms of synaptic convergence

The computational impact of input variation arises from the interplay of synaptic summation and the GBC's AP threshold. Like other auditory brainstem neurons, GBCs possess low input resistance and a fast membrane time constant, creating a short integration window that is well-suited for temporal processing [63–66]. Notably, our analysis revealed that the observed changes in firing rate and VS were uncorrelated with these intrinsic membrane properties. Instead, GBC output was primarily driven by the degree of input variation, allowing GBCs to operate along a functional continuum.

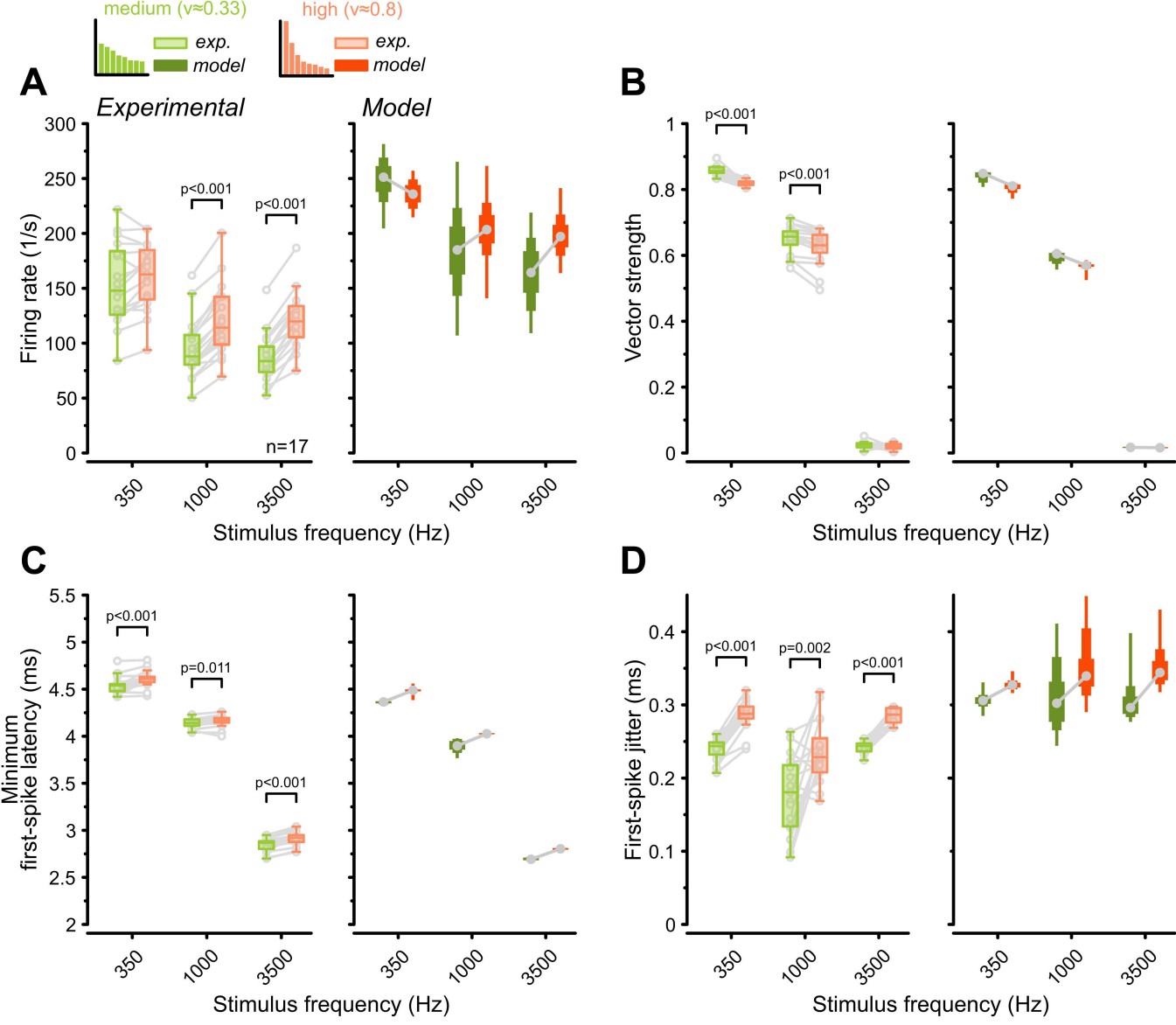

**Fig 5. Morphological input variation drives GBC functional heterogeneity.** Population data ($n = 17$) comparing responses to pure-tone stimulation under two input variation conditions derived from morphological estimates: "medium variation" (CV ≈ 0.33, green, GBC21) and "high variation" (CV ≈ 0.8, orange, GBC18L) [31]. Histogram illustrates the strength distribution of eight AN inputs used. **(A)** Left: High input variation enhanced GBC firing at 1,000 and 3,500 Hz, with no effect at 350 Hz. Right: GBC model responses corroborated the experimental results. **(B)** Temporal precision was reduced when GBCs received highly variable inputs. **(C, D)** Minimum first-spike latency was slightly longer at the "high variation" condition (C) and accompanied by an increase in first-spike jitter (D). Box plots containing experimental data show median and IQR (whiskers: 1.5 × IQR). Individual cells are shown as gray circles. Box plots for modeling data show the range of values from all 313 model instances with percentiles (thick: 25–75, medium: 10–90, thin: 0–100). *P*-values: 2-way RM ANOVA with Bonferroni post-hoc tests ($n = 17$). The data underlying this Figure are available at https://doi.org/10.17632/2tbkzdpjy6.

In the "no variation" scenario, where all inputs are uniformly subthreshold, AP generation requires precise temporal summation of multiple synaptic inputs. Consequently, GBCs act as coincidence detectors, filtering out uncorrelated input noise to achieve high temporal precision despite lower firing rates [36,61,67,68]. However, the capacity for input variation to enhance firing rates was frequency-dependent and more prominent during high-frequency stimulation (Fig 2B). At low

**Table 2. Summary of statistical analysis for morphologically-derived input variations.**

| | | Stimulus | morph$_{MED}$ | morph$_{HIGH}$ | | Post-hoc comparison (morph$_{MED}$ vs. morph$_{HIGH}$) |
|---|---|---|---|---|---|---|
| Fig 5 (*n*=17) | Firing Rate (Hz) | 350 Hz | 154±37 | 161±30 | p=3.7e−11 | p=0.05 |
| | | 1,000 Hz | 94±28 | 121±33 | | p=3.8e−9 |
| | | 3,500 Hz | 88±22 | 120±26 | | p=1.3e−10 |
| | Vector Strength | 350 Hz | 0.86±0.01 | 0.82±0.01 | p=2.6e−8 | p=2e−10 |
| | | 1,000 Hz | 0.65±0.04 | 0.62±0.05 | | p=5.3e−5 |
| | | 3,500 Hz | 0.02±0.01 | 0.02±0.01 | | p=0.41 |
| | Minimum first-spike latency (ms) | 350 Hz | 4.54±0.09 | 4.61±0.1 | p=2.2e−7 | p=2e−4 |
| | | 1,000 Hz | 4.14±0.05 | 4.16±0.07 | | p=0.01 |
| | | 3,500 Hz | 2.84±0.07 | 2.91±0.07 | | p=5.3e−7 |
| | First-spike jitter (ms) | 350 Hz | 0.24±0.01 | 0.29±0.02 | p=3.4e−7 | p=2.5e−10 |
| | | 1,000 Hz | 0.18±0.05 | 0.24±0.04 | | p=0.002 |
| | | 3,500 Hz | 0.24±0.01 | 0.29±0.01 | | p=7.7e−10 |
| Fig 6 (*n*=19) | Firing Rate (Hz) | 200 Hz | 134±30 | 142±27 | p=2.9e−10 | p=2e−5 |
| | | 600 Hz | 96±38 | 125±34 | | p=7.4e−9 |
| | | 2,000 Hz | 89±36 | 125±31 | | p=6.2e−11 |
| | Vector Strength | 200 Hz | 0.81±0.02 | 0.75±0.02 | p=3.8e−15 | p=2e−17 |
| | | 600 Hz | 0.46±0.04 | 0.37±0.02 | | p=8.1e−12 |
| | | 2,000 Hz | 0.01±0.01 | 0.01±0 | | p=0.07 |
| | Correlation index | 200 Hz | 2.78±0.11 | 2.33±0.1 | p=2.5e−18 | p=3.7e−19 |
| | | 600 Hz | 1.43±0.07 | 1.26±0.03 | | p=1.2e−10 |
| | | 2,000 Hz | 1±0.01 | 1±0 | | p=0.19 |
| | Corr$_{Norm}$ | 200 Hz | 0.89±0.01 | 0.93±0.01 | p=3.8e−11 | p=2.9e−17 |
| | | 600 Hz | 0.99±0.01 | 0.98±0 | | p=0.002 |
| | | 2,000 Hz | 0.81±0.01 | 0.82±0.01 | | p=0.04 |

frequencies, the highly synchronized activity of subthreshold inputs generates APs in most stimulation cycles (i.e., high degree of entrainment), effectively placing a ceiling on overall firing rates [36,67,69]. In contrast, the convergence of many desynchronized subthreshold inputs can raise the effective spike threshold, rendering the neuron more selective to highly correlated inputs during sensory encoding [70,71]. The principle of input convergence to enhance temporal precision is not limited to our experimental setting. For instance, experiments using optogenetic stimulation of the AN demonstrated that convergent activation improves temporal precision [72]. Similarly, computational models suggest that the loss of temporal precision in mice with impaired neurotransmitter release can be rescued by increasing synaptic convergence [73]. High input convergence has also been identified as a mechanism to improve temporal fidelity in the visual cortex and in electric fish [74,75].

In contrast, the "high variation" condition, characterized by a mixture of strong and weak inputs, shifts this operational mode by reducing the dependence on input coincidence. If the largest input is suprathreshold ("high variation" condition) it can independently drive the cell. Even when all inputs are subthreshold ("medium variation"), their combined activity depolarizes the membrane and introduces stochastic fluctuations that broaden the integration time window, thereby enhancing the neuron's responsiveness [27]. This shift in input configuration transforms GBCs into rate encoders, where the output is primarily dictated by the firing rate of the largest input. While a single large input may function similarly to multiple, perfectly synchronized small inputs, the noise-averaging benefits of convergence are lost. As a result, temporal jitter and stochasticity in the strong input translate directly into the GBC output, degrading temporal precision.

 

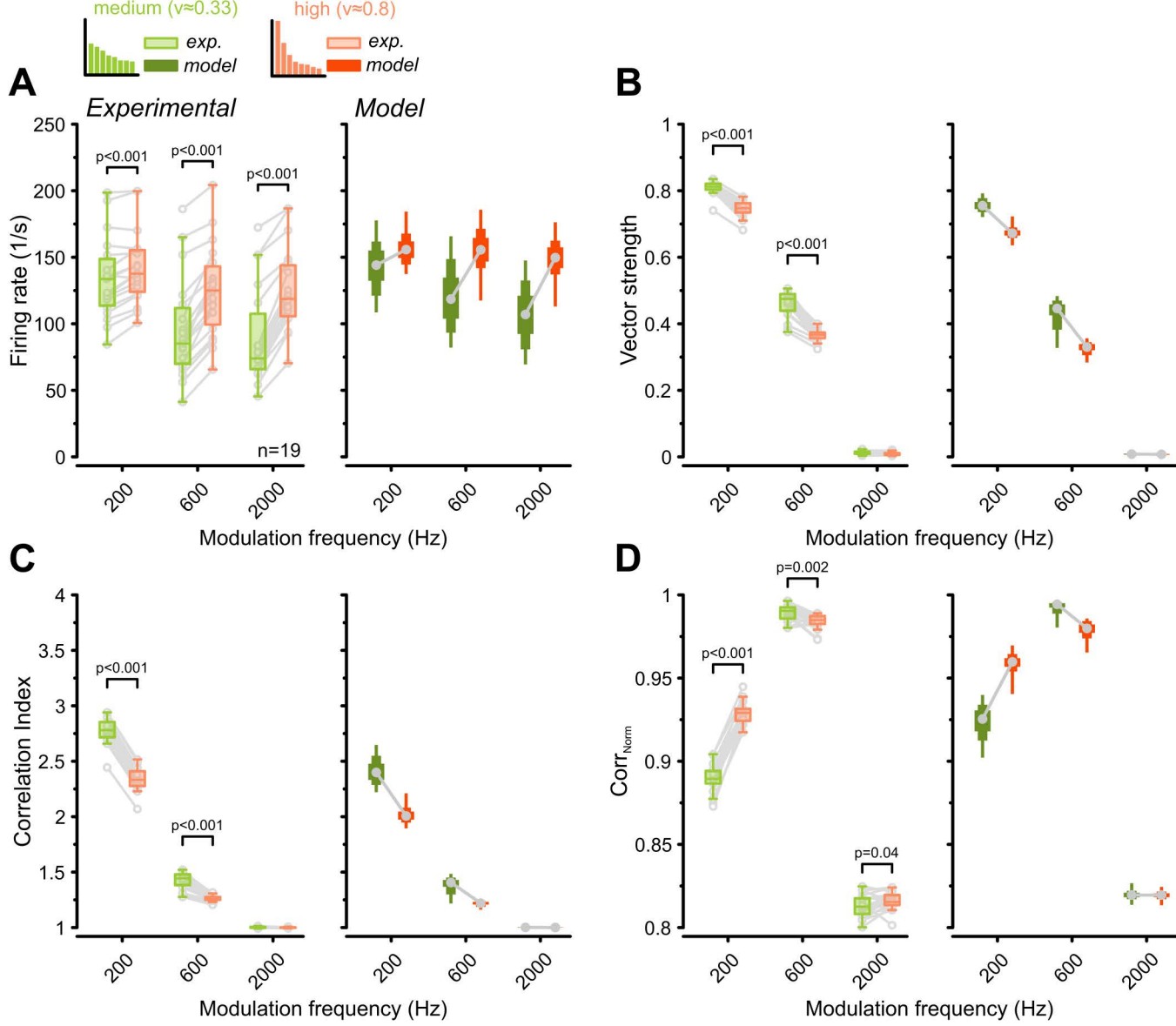

**Fig 6. Morphologically constrained input variation differentially affects temporal precision and envelope encoding of amplitude-modulated sounds.** Population data ($n = 19$) comparing GBC responses to SAM stimuli under two morphologically constrained input variations: "medium variation" (CV ≈ 0.33, green, GBC21) and "high variation" (CV ≈ 0.8, orange, GBC18L) [31]. **(A)** High input variation enhanced firing rates. **(B, C)** High input variation decreased vector strength **(B)** and correlation index **(C)**. **(D)** High input variation improved representation of the stimulus envelope (Corr$_{Norm}$). All results were successfully reproduced in the computational model. Box plots containing experimental data show median and IQR (whiskers: $1.5 \times$ IQR). Individual cells are shown as gray circles. Box plots for modeling data show the range of values from all 313 model instances with percentiles (thick: 25–75, medium: 10–90, thin: 0–100). P-values: 2-way RM ANOVA with Bonferroni post-hoc tests ($n = 19$). The data underlying this Figure are available at https://doi.org/10.17632/2tbkzdpjy6.

The presence of suprathreshold inputs also affects response latency. In theory, strong inputs shorten first-spike latency by triggering an AP without the need for coincidence. Under these conditions, GBCs would act as latency detectors, driven primarily by the input with the earliest arrival time. However, we observed that increased input variation also elevates spontaneous firing. This elevated baseline activity can obscure the first stimulus-evoked spike, making it difficult to detect

statistically against background noise. Indeed, while first spikes may occur earlier in the "high variation" condition, the minimum first-spike latency calculated as statistical deviation from the spontaneous activity ($FSL_{min}$) indicated slightly longer response latencies. Understanding these underlying mechanisms is critical, as alterations in GBC firing directly shape the input to subsequent stages of auditory processing.

## Functional implications for auditory processing

GBCs provide the primary excitatory input to the MNTB, where their large-diameter axons form the calyx of Held synaptic terminals. The calyx–MNTB synapse is a critical relay station [6,63,76–78], providing precisely-timed inhibition to several key nuclei within the superior olivary complex (SOC) involved in sound localization (e.g., medial superior olive, MSO; lateral superior olive, LSO) and temporal gap detection (superior paraolivary nucleus, SPN) [10,79–82].

Our findings, supported by simulations based on detailed EM reconstructions [31], suggest that the GBC population is functionally heterogenous, exhibiting substantial diversity driven by AN input variations. GBCs receiving similarly sized, subthreshold endbulb inputs may function as high-fidelity temporal relays, preserving and even enhancing the temporal precision required for binaural computations on a sub-millisecond timescale for sound localization [31,83–85]. Conversely, GBCs receiving highly variable inputs, characterized by one or two large suprathreshold endbulbs alongside smaller ones, might trade some temporal precision to emphasize a more robust rate code. This enhances their ability to track stimulus envelopes and results in shorter response latencies, but increases temporal jitter. Indeed, morphological data from a limited sample in mice, combined with computational modeling, suggest that roughly half of GBCs operated in a coincidence detection mode, while the other half showed a mixed-mode strategy [31]. Whether this distribution generalizes to the whole population or differs along the tonotopic axis and across species remains to be investigated.

The implications of GBC heterogeneity on downstream targets are intriguing. Their thick axons project predominantly to the contralateral MNTB, forming the canonical calyx of Held, but some collaterals project to the periolivary nuclei, the VNLL, and the LSO [47,86,87]. The MNTB provides inhibitory inputs to several downstream targets with different functional demands. For instance, the MSO requires precisely timed inputs for interaural time difference coding, whereas LSO and SPN may require inputs that are more dependent on sound intensity. The required functional diversity might originate in both the MNTB and its GBC inputs. Indeed, the calyx of Held exhibits considerable structural and functional heterogeneity, irrespective of tonotopic location. For example, MNTB neurons contacted by simple calyces typically show stronger EPSC depression, reflecting differences in release probability [88,89]. Notably, recent work demonstrated that this presynaptic diversity is mirrored by the intrinsic excitability of their postsynaptic target. MNTB neurons receiving simple calyces exhibit lower input resistance, faster membrane time constant, and reduced AP fidelity than those contacted by complex calyces [90]. Our current data suggests that input variation alone may be sufficient to shape GBC output, with changes in firing rate and VS being independent of intrinsic membrane properties. However, if presynaptic variation and postsynaptic properties were matched, as seen in the MNTB, this could further amplify the functional diversity among GBCs. It is tempting to speculate that this GBC diversity corresponds to the heterogeneity of their calyx terminals, enabling the MNTB to meet the specific computational demands of its target nuclei. Although direct evidence linking GBC input variation to specific calyx subtypes or SOC targets is currently lacking, a heterogeneous GBC population capable of encoding various stimulus features could diversify MNTB signaling to downstream nuclei.

## Biological origin of synaptic variation and methodological considerations

The biological origin of synaptic variation in GBCs remains unclear. One potential source of this heterogeneity is the convergence of different spiral ganglion neuron (SGN) subtypes onto individual GBCs. In mice, for example, BCs receive inputs from multiple SGN subtypes with distinct acoustic response properties [91,92]. In contrast, data from cats suggest that individual BCs are primarily contacted by a single SGN subtype, with type Ia endbulbs being larger than those from type Ic [32,93,94]. If gerbils follow the cat model and SGN subtypes systematically correlate with endbulb size, their

interaction during sound stimulation would introduce a heterogeneity not explicitly addressed in the current approach. In mice, however, SGN inputs onto the same BC exhibit comparable short-term plasticity [91,95,96], and BCs with varying SGN proportions display similar firing thresholds and SRs [91]. This suggests that intrinsic variations in synapse size, potentially independent of SGN subtype, are the primary drivers of the observed functional effects.

Beyond SGN subtype identity, heterogeneity in frequency tuning and developmental plasticity may also contribute to input variation. Physiological studies in cats indicate that some GBCs may integrate inputs from AN fibers tuned to slightly different frequencies [97,98]. If large and small inputs were tuned differently, this would affect their functional convergence during complex acoustic stimuli, such as frequency sweeps. Furthermore, it is unknown whether this variation reflects a random outcome of synapse elimination during development or a dynamic feature shaped by auditory experience. In both rodents and cats, endbulb size changes with sound exposure, demonstrating the potential for activity-dependent modulation [99–103]. Such plasticity could provide a mechanism for neuronal adaptation without requiring synapse elimination. These unresolved biological factors present specific limitations for our study. Our recordings were performed in Mongolian gerbils, a species for which detailed anatomical data on input convergence is currently lacking. Nevertheless, the consistent effect of input variation across GBCs suggests that the underlying mechanism is robust and largely independent of the exact number of endbulbs terminating on a single neuron. The ability of the model to replicate these effects further implies that this mechanism functions across species-specific differences in input number and variation. By constraining input variations to morphological data from mouse GBCs [31], we demonstrated that biologically plausible variations in input size are sufficient to induce the observed effects on GBC coding.

Our experimental and modeling approaches assumed a linear summation of statistically independent synaptic inputs. This assumption was chosen based on the observation that endbulb terminals are generally distributed across the GBC's somatic surface, likely minimizing interactions between them [31,38]. While this model captured the core phenomena and revealed a consistent effect across individual neurons, biological GBCs likely exhibit additional complexities, particularly in vivo. For example, multiple endbulbs originating from the same AN fiber or exhibiting highly correlated activity could function as a single, large input, thereby reducing the effective degree of variation. Additionally, morphological evidence suggests that larger inputs often terminate near the axon initial segment, potentially amplifying their synaptic efficacy beyond size alone [31]. Finally, while GBC dendrites are typically short, they may receive additional synaptic inputs that could modulate integration dynamics and excitability in a state-dependent manner [31,33,34,62,104,105]. Despite these simplifications, the consistent reproduction of key experimental findings in the computational model supports the conclusion that input variation is a key regulator of GBC output. Future studies employing large-scale electron microscopy reconstructions across multiple ages and CN locations could provide the necessary structural details to investigate how these effects are impacted by modulatory influence such as somatic and dendritic inhibition [106–113].

## Input heterogeneity as a general coding strategy

The integration of strong and weak synaptic inputs is a fundamental principle of neural circuit design, evident in the spinal cord and various brain regions, including the cortex and thalamus [22–26,114,115]. This organization allows neurons to perform complex computations and is found across multiple sensory systems, often beginning at the periphery. Synapses between the inner hair cells and the contacting SGN exhibit different calcium sensitivities and release probabilities. This arrangement creates parallel output channels from the same inner hair cell, each with distinct sensitivities and thresholds [116]. Similarly, in the cerebellum, granule cells are innervated by mossy fibers with varying strengths and short-term plasticity profiles, a heterogeneity thought to enable pathway-specific responses that improve temporal coding of multisensory inputs [117]. Cerebellar nuclei receive inhibitory inputs of highly variable sizes from Purkinje cells, allowing them to convey both rate codes and precise temporal information [28]. A similar principle of variable input strength is found in barrel cortex [27]. In each of these cases, similar to GBCs, structural heterogeneity expands the encoding capacity of the neural circuit.

At the network level, neuronal heterogeneity is considered essential for maximizing computational power. Theoretical and experimental work in the cortex has shown that diversity in both synaptic weights and intrinsic cellular properties is critical for robust stimulus encoding and learning [118–120]. Heterogeneous neuronal populations are more effective in learning tasks with complex temporal structures [121] and provide a richer, more reliable representation of sensory information [4,122].

Our findings place GBCs within this broader framework by illustrating how synaptic input variation can generate diverse output streams, some optimized for temporal fidelity and others optimized for rate. This heterogeneity might therefore be critical for tailoring information coding to the different computational demands of downstream auditory circuits.

## Methods

### Ethics statement

All experiments were conducted at the Physiology Laboratories of the Department for Human Medicine at the University of Oldenburg. Experimental protocols were approved by the University's Animal Welfare Office and carried out in accordance with the German Animal Welfare Act (§4 TSchG) and European Union Directive 2010/63/EU (protocol number: Milenkovic TschG 4 (3)). Every effort was made to minimize the number of animals used and to alleviate potential pain or distress.

### Animals

Mongolian gerbils (*Meriones unguiculatus*) were bred at the central animal facility of the University of Oldenburg. Animals were group-housed under a standard 12-h light/dark cycle with *ad libitum* access to food and water. Experiments were performed on young adults (postnatal day P20–P25) of both sexes ($n = 35$; 12 female).

### Method details

**Acute brain slice preparation.** Following rapid decapitation, acute parasagittal slices (200 μm thickness) containing the CN were prepared using a vibrating microtome (Leica VT1200 S) equipped with zirconia ceramic blades (EF-INZ10, Cadence Blades). Slicing was performed in ice-cold, low-calcium artificial cerebrospinal fluid (aCSF) containing (in mM): 125 NaCl, 2.5 KCl, 0.1 $CaCl_2$, 3 $MgCl_2$, 1.25 $NaH_2PO4$, 25 $NaHCO_3$, 10 glucose, 3 myo-inositol, 2 sodium pyruvate, and 0.4 ascorbic acid. The aCSF was continuously oxygenated with carbogen (5% $CO_2$ and 95% $O_2$) and maintained at pH 7.4. Slices were then transferred to a holding chamber containing standard recording solution at near-physiological temperature (36–37°C; identical to the slicing aCSF, except $CaCl_2$ and $MgCl_2$ were adjusted to 1.2 mM and 1 mM, respectively), allowed to recover for 30–45 min, and stored thereafter until recording. All solutions were prepared using ultrapure water (0.055 μS/cm).

**Electrophysiology.** Slices were placed in a recording chamber continuously perfused with oxygenated aCSF (~1 mL/min) and maintained at ~37°C using a dual automatic temperature controller (TC-344B, Warner Instruments). Neurons were visualized using an upright microscope (Axioskop 2 FS plus, Zeiss) with a water-immersion 63× objective (W Plan-Apochromat 63×/1.0, Cat# 421480-9900-000, Zeiss) and a CCD camera (IR-1000, DAGE-MTI).

Recording pipettes were pulled from borosilicate glass (GB-150F-8P, Science Products) to a resistance of 3–5 MΩ when filled with intracellular solution containing (in mM): 120 K-gluconate, 20 KCl, 5 EGTA, 10 HEPES, 0.05 $CaCl_2$, 5 $Na_2$-phosphocreatine, 4 Mg-ATP, and 0.3 Na-GTP, supplemented with 0.2% biocytin for post-hoc morphological verification. Voltages were corrected for a calculated liquid-junction potential of 12 mV. Whole-cell recordings were made using a Multiclamp 700B amplifier (Molecular Devices), digitized at a 100 kHz sampling (Digidata 1440A interface, Molecular Devices) and acquired using pClamp software (RRID:SCR_011323, version 11.2.2, Molecular Devices). In current-clamp mode, resting membrane potential was held near −72 mV using a small bias current (<140 pA). Series resistance (<10 MΩ) was fully compensated via bridge balance. Pipette capacitance neutralization and bridge balance were monitored

and adjusted throughout. Intrinsic membrane properties were characterized for a subset of GBCs (70/83, 84%) using a 200 ms square current pulse (50–100 pA).

**Simulating auditory nerve fiber input.** Spiking activity of AN fibers was simulated using the 2018 version of the auditory periphery model [40] in the "cat" configuration, as gerbil AN fiber responses resemble those of cats [123–125]. Since GBCs primarily receive inputs from high SR type Ia SGN [32,91,92], model fiber SR was set to 70 spikes/s. Refractory periods were fixed at the midpoint of the model's default parameter range, representing typical physiological values (absolute: 0.45 ms, relative: 0.5125 ms).

Consistent with anatomical evidence that GBCs receive around 5–12 endbulb inputs in mice and around 20 in cats [31,34], we simulated the convergence of 10 AN inputs per GBC. Given the narrow frequency tuning of GBCs [69], all converging simulated AN fibers shared the same characteristic frequency (CF). For pure-tone stimuli, the fibers' CF matched the tone frequency; for amplitude-modulated (AM) sounds, the CF matched the carrier frequency. Sound intensity was 70 dB SPL for pure tones and 25 dB SPL for AM sounds. Tone stimuli were simulated for 200–500 ms with an identical pause between repetitions to mimic stimulation paradigms used during in vivo recordings [107,126–128]. The AN spike pattern for a given stimulus remained fixed across cells and variation conditions.

**Controlling synaptic input variation.** To investigate the impact of input heterogeneity, we systematically varied the strength of the 10 simulated AN fibers. Synaptic input variation ($v$) was quantified as the coefficient of variation, defined as the standard deviation ($s$) of the 10 AN input amplitudes divided by their mean ($m$; i.e., $v = s/m$). We used a gamma distribution (shape parameter $a = 1/v^2$, scale parameter $b = mv^2$) to create three conditions: "no variation" ($v = 0$), "medium variation" ($v = 0.4$), and "high variation" ($v = 1$) (Fig 1B). These levels cover a broad range of synaptic size heterogeneity, including the range reported from anatomical estimates of GBC inputs [31].

To isolate the effect of input variation, average input strength was first adjusted to 7–14 nS to elicit GBC firing rates of >100 spikes/s during 350 Hz stimulation. This average input strength was held constant across conditions. In the "no variation" condition, all inputs had equal strength and were functionally subthreshold. The "medium variation" condition featured inputs ranging from 3–6 nS (smallest input) to 12.5–25 nS (largest input) with all inputs remaining subthreshold. In the "high variation" condition, inputs ranged from 0.2–0.4 nS (smallest) to 23–46 nS (largest), with one suprathreshold input. Total charge between conditions was nearly identical, with the small difference (<2%) reflecting the stochasticity of AN fiber discharges. Stimuli were presented sequentially while monitoring cell vitality to exclude time-dependent confounds.

To confirm the biological relevance of our findings, we also implemented synaptic strength distributions based on the morphological data [31]. Two representative GBC input data sets were selected: one previously characterized as fitting the coincidence detection model (GBC21, CV ≈ 0.33, approximating our "medium variation," morph$_{MED}$) and another fitting the mixed-mode integration model (GBC18L, CV ≈ 0.8, approximating our "high variation," morph$_{HIGH}$). To facilitate comparison between these two morphologically derived input variations, we used eight AN inputs for both conditions.

**Conductance-clamp implementation.** Synaptic conductance traces ($G_{syn}$) were generated by convolving simulated AN spike trains with a stereotypical EPSC waveform, modeled as alpha-function with a time constant of 0.2 ms. EPSC amplitudes were scaled according to the input variation condition. Total $G_{syn}$ was calculated as the linear sum of conductances from all AN fibers.

Conductance-clamp was implemented as described previously [129]. Briefly, the instantaneous GBC membrane potential ($V_m$) was digitized (PCIe-6361 via BNC-2120 interface, National Instruments) and fed into a separate computer running mafDC (courtesy of M. A. Xu-Friedman) in Igor Pro (RRID:SCR_000325, version 8, Wavemetrics). The appropriate AMPA current ($I_{syn}$) was calculated in real-time (latency: 15 μs, AMPAR cycle time: 8 μs), according to Ohm's law, assuming a reversal potential ($E_{syn}$) of −12 mV, and injected back into the GBC via the recording electrode:

$$I_{syn} = G_{syn} (V_m - E_{syn}).$$

**Post-hoc morphological identification of GBCs.** To morphologically confirm GBC identity, slices with biocytin-filled neurons were fixed with 4% paraformaldehyde in 1× phosphate-buffered saline (PBS) overnight at 4°C. Following fixation, slices were rinsed in 1× PBS (6×5 min) and 1× PBS with 0.3% TritonX-100 (PBST, 6×5 min). To visualize biocytin-filled GBCs, slices were incubated for 2.5 h at room temperature with Cy5-conjugated streptavidin (5 μg/mL in PBST, Cat#016-170-084, Jackson ImmunoResearch). After final rinses with PBST (6×5 min), PBS (2×5 min), and ultrapure water (2×5 min) slices were mounted on glass slides (Aqua-Poly/Mount, Cat#18606, Polysciences). Confocal images were acquired on a Leica SP8 confocal scanning microscope equipped with a 10× (0.3 NA, Cat#11506505) dry objective and a 63× (1.4 NA, Cat#11506350) oil-immersion objective using a 647 nm excitation wavelength.

**Globular bushy cell model.** Computational simulations of GBC responses were performed using a modified version of the "adaptive coincidence-counting model" [4]. Each simulated AN spike evoked a simplified rectangular postsynaptic response (Fig 2A) of constant duration ($W$) but variable amplitude ($A$) depending on the variation condition. These responses were linearly summed to produce the total integrated input $u(t)$. The model generated an output spike when $u(t)$ exceeded an adaptive threshold $\theta(t)$, followed by an absolute refractory period ($R$) during which no further spikes could be triggered.

The adaptive threshold consisted of static and dynamic components: $\theta(t) = \theta_S + \theta_D(t)$. The static component $\theta_S$ was fixed at 1, so that postsynaptic response amplitudes ($A$) were considered normalized to this static input threshold. The dynamic component $\theta_D$ changed according to the total input amplitude: $T(d\theta_D/dt) = -\theta_D(t) + S \cdot (u(t))^2$, where $T$ is the time constant and $S$ is the strength of adaptation [4]. The quadratic exponent of $u(t)$ was designed to approximate the nonlinear activation of various adaptive mechanisms in BC, such as low-threshold K⁺ channels and inhibitory feedback from other CN neurons. This adaptive threshold mechanism allowed the model to reject poorly synchronized inputs (Fig 2A). The spike generation mechanism implemented in the GBC model is deterministic and does not produce spike time jitter. To simulate this intrinsic variability, a spike-generation delay $D_{spike} = D_{fix} + D_{rand}$ was added to each output spike timing. The fixed component $D_{fix}$ was set to 0.2 ms, and the random component $D_{rand}$ was drawn from an alpha distribution with a time constant of 0.1 ms [130]. This modification was based on the spike latency data from pure-tone stimulation (Fig 3B5 and 3B6) and increased response latency by 0.4 ms (SD = 0.14 ms).

**Model parameters and validation.** The GBC model included five free parameters (Table 3). A default set was established to match experimental pure-tone responses (e.g., PSTHs and raster plots in Fig 3). To confirm the robustness of the modeling results, we systematically varied parameters around this default (Table 3, right column). Each "model instance," representing a unique parameter combination, was tested with pure-tone stimulation. Only those instances exhibiting GBC-like responses, as defined by the quantitative criteria described below, were adopted for further simulations.

The adoption criteria were based on our experimental observations (Figs 1 and 3). Model instances were required to satisfy the following criteria: (1) SR of 10–60 spikes/s; (2) for 350 Hz stimulation: steady-state driven rate >180 spikes/s and VS >0.85; (3) for 3,500 Hz stimulation: steady-state driven rate of 80–180 spikes/s and onset spike probability >0.95. Furthermore, the PSTH shape near the onset of the 3,500 Hz tone was examined to ensure a primary-like-with-notch (PL_N) spiking pattern, characterized by a sharp peak followed by a notch (pause of spiking activity) and a smooth recovery towards

**Table 3. Model parameters. The total number of parameter combinations was 12×7×7×7×7 = 28,812. The default parameter values were used for Figs 3A3, 3C1, 4A3.**

| Parameter | Default value | Ranges tested |
|---|---|---|
| (mean) Input amplitude $A$ | 0.38 | 0.28–0.50 (step 0.02) |
| Input duration (ms) $W$ | 0.6 | 0.2–1.4 (step 0.2) |
| Refractory period (ms) $R$ | 1.5 | 0.8–2.0 (step 0.2) |
| Adaptation strength $S$ | 0.8 | 0.4–1.6 (step 0.2) |
| Adaptation time constant (ms) $T$ | 0.7 | 0.3–2.1 (step 0.3) |

the steady-state spiking response (Fig 3A2). Specific criteria for the $PL_N$ response were: (4) the notch appeared within 3 ms after the onset peak; (5) the notch amplitude (minimum rate in PSTH) was below 50% of the steady-state spike rate; (6) a second peak in the PSTH was absent or below 300 spikes/s; (7) the average spike rate at 6–10 ms after the stimulus onset was > 90% of the steady-state spike rate; and (8) a second notch was absent or higher than 50% of the steady-state spike rate. The last criterion was used to distinguish $PL_N$-type PSTHs from chopper-type PSTHs [4]. Of the 28,812 model instances tested, 313 (1.1%) met all of these physiological response criteria to be considered "GBC" and were used for simulations.

## Quantification and statistical analysis

Individual GBCs were treated as independent biological samples. Aggregated data are presented as mean ± standard deviation, unless otherwise noted. All comparisons involved within-cell manipulations (i.e., stimulus type and input variation for the same cell) and were analyzed using a multifactorial repeated-measures ANOVA (two-way RM ANOVA). Post-hoc multiple comparisons were corrected using the Bonferroni method [131]. A $p$-value <0.05 was deemed statistically significant for all interpretations. As no sex difference was found for firing rate ($p = 0.25$) or temporal precision ($p = 0.86$), data were pooled for further analysis. Sample sizes were determined based on variance observed in similar studies. No statistical methods were used to pre-determine sample sizes prior to data collection.

**Electrophysiological data analysis.** Data analysis was performed using custom-written scripts in MATLAB (RRID:SCR_001622, Version 2024b, The Mathworks). Input resistance was derived from a linear fit to the current-voltage relationship in response to a square current pulse, while the membrane time constant was estimated by fitting a single-exponential function to the initial phase of the voltage deflection. APs were detected at −20 mV threshold, and all detected spikes were visually inspected and curated to exclude potential artifacts. For all subsequent analyses, the time of the AP peak was used as the spike time. Average firing rates were calculated for evoked (200–500 ms duration) and spontaneous (150 ms duration) epochs. Correlations were assessed using Spearman's rank [132]. Onset probability was calculated as the fraction of trials with at least one AP within 1.5 ms of the response onset. Minimum first-spike latency ($FSL_{min}$) was defined as the time between stimulus onset and the detection of a statistically significant firing rate increase ($p < 1e−6$) above the SR [49]. This method avoids the ambiguity in identifying the first stimulus-evoked spike caused by varying spontaneous firing rates. First-spike jitter was quantified as the standard deviation of spike latencies occurring within a window from 0.6 ms before and 1.2 ms after $FSL_{min}$. PSTH were constructed using 0.5 ms bin width across 100–200 repetitions. Phase-locking to periodic stimuli (pure tones or SAM) was quantified by VS [41]. Spike timing precision was assessed by CI, and calculated as the normalized peak height of the shuffled autocorrelogram, constructed across repeated presentations of identical stimuli [58,59]. Envelope tracking for SAM stimuli was quantified by cross-correlation ($Corr_{Norm}$) between the stimulus envelope and the corresponding response PSTH, after correcting for the neuron's response latency [106,133]. Note that $Corr_{Norm}$ values range from 0 to 1; unmodulated responses typically yield a $Corr_{Norm}$ value around 0.82, while values >0.9 indicate robust tracking of the stimulus envelope. Entrainment index was used to quantify the proportion of interspike intervals within 0.5–1.5 times the stimulus period [36].

## Acknowledgments

The authors acknowledge Mathias Dietz for helpful discussions and the Fluorescence Microscopy Service Unit at the Carl von Ossietzky University of Oldenburg for the use of the imaging facilities.

## Author contributions

**Conceptualization:** Go Ashida, Christian Keine, Ivan Milenkovic.

**Data curation:** Go Ashida, Christian Keine.

**Formal analysis:** Go Ashida, Christian Keine.

**Funding acquisition:** Christian Keine.

**Investigation:** Chunjian Wang, Go Ashida.

**Methodology:** Go Ashida, Christian Keine.

**Project administration:** Ivan Milenkovic.

**Resources:** Go Ashida, Ivan Milenkovic.

**Software:** Go Ashida, Christian Keine.

**Supervision:** Christian Keine, Ivan Milenkovic.

**Validation:** Go Ashida.

**Visualization:** Christian Keine.

**Writing – original draft:** Christian Keine.

**Writing – review & editing:** Go Ashida, Christian Keine, Ivan Milenkovic.

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
