## [Editor Report · Decision Letter 0]

15 Sep 2025

Dear Dr Keine,

Thank you for submitting your manuscript entitled "Variation in Synaptic Inputs Drives Functional Versatility for Sound Encoding in Globular Bushy Cells" for consideration as a Research Article by PLOS Biology.

Your manuscript has now been evaluated by the PLOS Biology editorial staff, as well as by an academic editor with relevant expertise, and I am writing to let you know that we would like to send your submission out for external peer review.

Once your full submission is complete, your paper will undergo a series of checks in preparation for peer review. After your manuscript has passed the checks it will be sent out for review. To provide the metadata for your submission, please Login to Editorial Manager (https://www.editorialmanager.com/pbiology) within two working days, i.e. by Sep 17 2025 11:59PM.

Kind regards,

Taylor

Taylor Hart, PhD,

Associate Editor

PLOS Biology

thart@plos.org

---

## [Decision Letter · Decision Letter 1]

21 Oct 2025

Dear Dr Keine,

Thank you for your patience while your manuscript "Variation in Synaptic Inputs Drives Functional Versatility for Sound Encoding in Globular Bushy Cells " went through peer-review at PLOS Biology. Your manuscript has now been evaluated by the PLOS Biology editors, an Academic Editor with relevant expertise, and by several independent reviewers.

In light of the reviews, which you will find at the end of this email, we are pleased to offer you the opportunity to address the comments from the reviewers in a revision that we anticipate should not take you very long. We will then assess your revised manuscript and your response to the reviewers' comments with our Academic Editor aiming to avoid further rounds of peer-review, although we might need to consult with the reviewers, depending on the nature of the revisions.

**IMPORTANT - SUBMITTING YOUR REVISION**

*Resubmission Checklist*

*Published Peer Review*

*PLOS Data Policy*

*Blot and Gel Data Policy*

Sincerely,

Taylor

Taylor Hart, PhD,

Associate Editor

PLOS Biology

thart@plos.org

REVIEWS:

Reviewer #1: Summary

This manuscript combines technically demanding in vitro conductance-clamp recordings with computational modeling to investigate how heterogeneity in auditory nerve input strength affects signal encoding in globular bushy cells (GBCs). The authors demonstrate that increasing input variability enhances firing rate but reduces temporal precision, thereby placing GBCs along a functional continuum between coincidence detection and rate coding.

The topic is of high relevance to auditory neuroscience and provides a clear conceptual advance in understanding how morphological variability translates into computational diversity. The study is well designed, the results are internally consistent, and the combination of experimental and modeling approaches is compelling.

Overall, this is a solid and polished study that is nearly ready for publication pending minor revisions aimed at improving language, precision, and contextual discussion.

Major Comments

1. Contextualization with prior work:

The discussion would benefit from incorporating additional literature demonstrating convergence-related enhancement of temporal precision. Two relevant studies the authors could consider are:

o Mittring et al., 2023, Brain Stimulation — Showing that convergent optogenetic stimulation of auditory fibers improves temporal precision in bionic sound encoding.

https://www.brainstimjrnl.com/article/S1935-861X(23)01673-X/fulltext

o Strenzke et al., 2009, J. Neurosci. — Demonstrating impaired timing fidelity with reduced presynaptic release (complexin 1 mutation), supporting convergence as a mechanism to improve temporal coding.

https://www.jneurosci.org/content/29/25/7991

Including these will help position the study within a broader context of how convergence and presynaptic mechanisms jointly shape temporal precision in auditory processing.

2. Physiological interpretation of input variability:

The Discussion could more explicitly address potential biological origins of the observed heterogeneity (e.g., developmental variability, experience-dependent plasticity, or differences between low- and high-frequency GBCs).

3. Language and clarity:

While the manuscript reads well, some sentences—especially in the Introduction (lines 51-74) and Discussion—are overly long and could be streamlined for clarity. For example, repeated formulations like "input variation shapes GBC output" could be replaced with more specific phrasing ("increased variance in excitatory conductances enhances firing rate but reduces vector strength").

4. Terminological precision:

Define "temporal precision" early in the Results section in relation to the quantitative metrics used (vector strength, correlation index). Also clarify when "high variation" implies inclusion of suprathreshold inputs versus subthreshold ones.

Minor Comments

* Figures and legends: Please restate sample sizes (n) and the statistical test used in each figure legend for transparency.

* Abstract: Consider adding one sentence that summarizes the key finding and its trade-off, e.g. "Input heterogeneity enhances rate coding at the expense of temporal precision."

* Discussion: When referring to downstream processing, briefly note implications for binaural nuclei such as MSO, where BC output patterns are functionally critical.

* Style and consistency: Maintain consistent tense throughout Methods and Results. Verify that all data and model repository links are accessible.

Strengths

* Excellent integration of electrophysiology and modeling, providing a mechanistic link between structure and function.

* Clear conceptual advance on how input heterogeneity contributes to computational versatility in the auditory brainstem.

* Strong alignment between experimental and modeled results.

* Relevance to broader issues of rate vs. temporal coding in sensory systems.

Weaknesses

* Occasional lack of terminological precision.

* Some overly long or repetitive phrasing that slightly hinders readability.

Reviewer #2: This study examined the impact of the diversity of axonal input strength at auditory nerve - globular bushy cell synapses in gerbil cochlear nucleus in vitro. Based on previous datasets, the authors designed dynamic clamp protocols to simulate realistic activity of these synapses while monitoring output spike probability and timing. The primary conclusion is that when the population of synapses a cell receives varies in strength, the neuron will have an enhanced firing probability (sustained driven rate and spont rate) but at the cost of timing (increased jitter). The results are straightforward and I have only few concerns about the approach and the writing. Overall I would describe the work as a logical extension of the EM/modeling work from Spirou and Manis, but one that brings their conclusions to a more physiological setting.

Specific comments:

Missing information on stimulus paradigm:

1. What is the strength of the single inputs modeled for each of the three input variations conditions? (Fig. 1B plots the ANF strength vs the AND # without scaling. Providing the numbers is essential for reproducibility, thus they should at least scale the y-axis in this figure). It wasn't entirely clear how the values for variation were chosen.

2. Is the total input strength (i.e., the sum of the ten input) the same across the three conditions? (should be similar, according to the gamma distribution from which the unitary input strength are drawn, but not necessarily the same).

3. Is the stochastic nature of the output solely due to the stimulus tokens or also to that of the postsynaptic properties?

4. Line 479 authors write that they adjust the mean input strength to a level that elicits a firing rate of about 100 spikes/s in response to a 350 Hz (sound frequency) stimulus. Providing information about these numbers (mean input strength) would allow one to get an idea about the excitability of the neurons, and the variability in excitability. Also, this is an important measure for to enable reproducibility of the study. (this is different in the model, where the mean input strength is a parameter, which they adjusted to match the conductance-clamp data)

5. Authors present each stimulus condition 15 times (e.g., Fig.3B1). Are the conductance stimuli the same 15 across the neurons, or is the conductance stimulus created for each experiment separately?

6. At which inter stimulus interval are the conductance stimuli presented?

7. Not too important, but: 3 input variations x 15 repeats = 45 stimulations per neuron. Are these 45 stimulations presented in order or are they randomized?

8. Fig. 1C1 (3500 Hz) and Fig. 3B3 show results for the same stimulus, but firing rates that are about 25 - 40 % lower in Fig. 3B3 (calculated from the data in Table 2). Why?

Discussion:

1. Authors put a lot of emphasis on the contribution of the input variation to temporal and rate coding, but do not consider differences in the excitability of the neurons. Actually, they take steps to reduce differences due to differences in excitability of the neurons (adjusting mean strength of the inputs to match 100 spikes/s). In other words: Can a neuron be tuned to enhance temporal coding by its intrinsic properties (in respect to the GBC population and a given input variation).

2. Overall the work has a fairly descriptive feel to it: a set of manipulations were done, the results compared, yet while conclusions describe the results, there is no intuitive or biophysical explanation or understanding of why, for example, diversity leads to increased rate and more jitter. It's ironic considering that biophysical analyses are applied here. In this regard the work was a bit uninstructive.

3. Input variation is described as a means to optimize information transfer in the population. Perhaps it is selected for, but it might also be an epiphenomenon of synapse elimination during development, which leads to reduction of some inputs and not others.

4. Authors state that half the GBCs operate as coincidence detectors and the other half shows mix-mode operation, citing Spirou. Of course, Spirou's EM paper examined only a tiny fraction of cells and we should not speculate about what the population is like.

Reviewer #3: This manuscript investigates how the distribution of synaptic strengths onto a type of cell that operates mainly as a temporal coincidence detector influences the temporal patterns of spike generation and the rate of spontaneous activity. Specifically, the work uses a combination of brain slice electrophysiology, with dynamic clamp, and a reduced computational model, to explore this question in the bushy neurons of the gerbil cochlear nucleus. The conclusions are that: 1) when all inputs are sub-threshold and of similar strength, The modeling and experimental data are largely in agreement (there are some exceptions), suggesting that the model captures key mechanisms that control the cell's spiking patterns.

The question is an important and interesting one. The work is inspired by previous anatomical and modeling studies in the cochlear nucleus that have assayed the input terminal sizes (as a proxy for synaptic strength), and recapitulates several of those results, as well as modeling studies that examined combinations of fixed levels of sub-threshold and supra-threshold inputs (Rothman et al 1993). However, it differs from those studies by using statistically-defined nearly continuous distributions of synaptic strength, rather than either uniform synapses or morphologically-specified synapses from single-cell reconstructions. Importantly, by using dynamic clamp to test the consequences of different input patterns, this work provides a ground truth test of both the reduced model and of some prior modeling predictions. The work uses a reasonable selection of values for the parameter space, and while primarily focussed on 3 domains, results from intermediate distributions are also shown.

The overall conclusion is that the mixing pattern of sub-threshold inputs significantly affects the firing rates. A population bias towards large inputs increases both spontaneous and driven rates, first spike latency jitter, AM rate, and entrainment, while decreasing phase locking synchrony, the correlation index, SAM phase locking.

The paper is very clearly written, well organized and the figures are generally clear.

Minor:

line 165: The reference to Ritz and Brownell here seems inappropriate. That study focussed on the PVCN (mostly multipolar and octopus cells), and not on GBCs. Perhaps a different paper was intended?

line 179: Did you compute expectation for these "early" spikes based on the spontaneous rate, e.g., using the metric from Chase and Young? This would statistically separate potential spontaneous spikes from spikes that are "first spikes" based on spontaneous rate.

465: Just a comment: the idea that the convergence across frequency is limited is also supported to some extend by the Huffman sequence experiments of Carney (1990).

Methods; Mention any correction for the junction potential in current clamp, if applied. Otherwise, state "no correction"

Figures:

This is really minor, but the unequal widths of the panels in Figures 5 and 6 is a bit confusing. It would make more sense for the model panels to have the same horizontal space as the experimental panels.

---

## [Editor Report · Decision Letter 2]

9 Dec 2025

Dear Dr Keine,

Thank you for your patience while we considered your revised manuscript "Variation in Synaptic Inputs Drives Functional Versatility for Sound Encoding in Globular Bushy Cells" for publication as a Research Article at PLOS Biology. This revised version of your manuscript has been evaluated by the PLOS Biology editors and the Academic Editor.

Based on our Academic Editor's assessment of your revision, we are likely to accept this manuscript for publication (see below my signature for a comment from the Academic Editor). Please also make sure to address the following data and other policy-related requests.

IMPORTANT: Please ensure that you address these editorial requirements:

------------------

**Title:

We would like to suggest an alternate title for your paper, to emphasize the main findings and avoid a possible implication that they are based on in vivo data. Is this version of the title acceptable to you? We could also consider other suggestions that you might have.

"Variation in Synaptic Inputs Drives a Trade-Off Between Rate Encoding and Temporal Precision in Cochlear Nucleus Neurons"

**Ethics:

-- The Ethics statement needs to be a separate, independent (and the first) subheading in the Material & Methods section. It must include the full name of the IACUC/ethics committee that reviewed and approved the animal care and use, as well as the protocol/permit/project license number. https://journals.plos.org/plosbiology/s/ethical-publishing-practice

**Data

-- Please cite the location of the data clearly in all relevant Figure legends, e.g. “The data underlying this Figure can be found in https://doi.org/10.5281/zenodo.XXXXX”

------------------

We expect to receive your revised manuscript within two weeks.

*Published Peer Review History*

*Press*

Sincerely,

Taylor

Taylor Hart, PhD,

Associate Editor

thart@plos.org

PLOS Biology

COMMENT FROM THE ACADEMIC EDITOR (lightly edited):

I have now carefully looked at the revision and I think the authors have done an excellent job. I don't think another round of reviews is necessary and the paper can be accepted as it is

---

## [Editor Report · Decision Letter 3]

17 Dec 2025

Dear Dr Keine,

Thank you for the submission of your revised Research Article "Synaptic Input Variation Enhances Rate Coding at the Expense of Temporal Precision in Cochlear Nucleus Neurons" for publication in PLOS Biology. On behalf of my colleagues and the Academic Editor, Manuel Malmierca, I am pleased to say that we can in principle accept your manuscript for publication, provided you address any remaining formatting and reporting issues. These will be detailed in an email you should receive within 2-3 business days from our colleagues in the journal operations team; no action is required from you until then. Please note that we will not be able to formally accept your manuscript and schedule it for publication until you have completed any requested changes.

PRESS

Sincerely,

Taylor

Taylor Hart, PhD,

Associate Editor

PLOS Biology

thart@plos.org